# Low Hemoglobin Levels Are Associated with Reduced Psychomotor and Language Abilities in Young Ugandan Children

**DOI:** 10.3390/nu14071452

**Published:** 2022-03-30

**Authors:** Margaret Nampijja, Agnes M. Mutua, Alison M. Elliott, John Muthii Muriuki, Amina Abubakar, Emily L. Webb, Sarah H. Atkinson

**Affiliations:** 1Maternal and Child Wellbeing (MCW) Unit, African Population and Health Research Center, Nairobi 00100, Kenya; maggie.nampijja@gmail.com; 2Medical Research Council, Uganda Virus Research Institute and London School of Hygiene and Tropical Medicine Uganda Research Unit, Entebbe P.O. Box 49, Uganda; alison.elliott@lshtm.ac.uk; 3Kenya Medical Research Institute (KEMRI) Wellcome Trust Research Programme, KEMRI Centre for Geographic Medicine Research-Coast, Kilifi 230-80108, Kenya; jmuriuki@kemri-wellcome.org (J.M.M.); aabubakar@kemri-wellcome.org (A.A.); 4Department of Clinical Research, London School of Hygiene and Tropical Medicine, London WC1E 7HT, UK; 5Department of Public Health, School of Human and Health Sciences, Pwani University, Kilifi 195-80108, Kenya; 6Department of Psychiatry, University of Oxford, Oxford OX3 7JX, UK; 7Institute for Human Development, Aga Khan University, Nairobi 30270-00100, Kenya; 8MRC International Statistics and Epidemiology Group, Department of Infectious Disease Epidemiology, London School of Hygiene and Tropical Medicine, London WC1E 7HT, UK; emily.webb@lshtm.ac.uk; 9Centre for Tropical Medicine and Global Health, Nuffield Department of Medicine, University of Oxford, Oxford OX3 7FZ, UK; 10Department of Paediatrics, University of Oxford, Oxford OX3 9DU, UK

**Keywords:** anemia, iron, motor, cognitive, language, maternal, child, development, Africa

## Abstract

Children living in Sub-Saharan Africa are vulnerable to developmental delay, particularly in the critical first five years due to various adverse exposures including disease and nutritional deficiencies. Anemia and iron deficiency (ID) are highly prevalent in pregnant mothers and young children and are implicated in abnormal brain development. However, available evidence on the association between anemia, ID and neurodevelopment in sub-Saharan Africa is limited. Using data from the Entebbe Mother and Baby Study prospective birth cohort, we examined the effect of maternal and child hemoglobin (Hb) levels and child iron status on developmental scores in 933 and 530 pre-school Ugandan children respectively. Associations between Hb levels, iron status and developmental scores were assessed using regression analyses adjusting for potential confounders. Lower maternal and child Hb levels were associated with reduced psychomotor scores at 15 months, while only lower Hb levels in infancy were associated with reduced language scores. We found no evidence that anemia or ID was associated with cognitive or motor scores at five years. This study emphasizes the importance of managing anemia in pregnancy and infancy and highlights the need for further studies on the effects of anemia and ID in children living in Sub-Saharan Africa.

## 1. Introduction

About 81 million pre-school children in low- and middle-income countries have delayed cognitive and socioemotional development and 43.8% of these children live in Sub-Saharan Africa [1]. African children are vulnerable to developmental delay due to exposure to various risk factors including inadequately stimulating home environments, infections, malnutrition and micronutrient deficiencies [1,2]. Iron deficiency (ID) accounts for over 40% of all anemia cases and children living in Sub-Saharan Africa have the highest burden of ID and anemia globally [3,4]. It is estimated that approximately 60% of pre-school children living in Africa are anemic and 52% are iron deficient [3,5]. Additionally, 38% of pregnant women are anemic (Hb < 11 g/dL) globally and of these women, 46.3% live in Africa [6,7].

Animal and in vitro studies suggest that iron may influence neurobehavioral outcomes through its role in brain development. Studies report that iron is important for neurotransmitter synthesis and function, myelin formation, and DNA synthesis and repair in the brain [8,9,10]. In animal studies, ID is linked to impaired learning, memory function and behavioural changes which may be reversed by iron supplementation [11,12,13], however in other studies, the detrimental effects of ID are long lasting and cannot be reversed with iron supplementation [14,15]. Despite the high prevalence of anemia and ID in Sub-Saharan Africa, few studies have investigated its impact on early developmental outcomes and findings from these studies were mixed (Appendix A). There is also limited evidence on the longitudinal effects of anemia and ID prenatally and in early childhood on neurobehavioral outcomes assessed at different timepoints in pre-school African children. Only two cohort studies have evaluated the relationship between Hb, iron status and neurobehavioral outcomes in young African children. One multi-country study reported associations between child Hb, iron status and motor and language development in children at 18 months but reported no associations between maternal Hb, iron status and child development [16]. In one cohort, an inverted U-shaped association was reported between maternal Hb during pregnancy and motor development at one year, but maternal iron status was not associated with child development [17,18].

Using data from a large longitudinal, prospective birth cohort study, the Entebbe Mother and Baby Study (EMaBS), our study addressed three aims assessing: (1) the effect of Hb levels during pregnancy and early childhood on developmental scores at 15 months; and (2) at five years of age; and (3) the effect of ID and IDA at two years of age on developmental scores at five years of age.

## 2. Materials and Methods

### 2.1. Study Design and Setting

The Entebbe Mother and Baby Study (EMaBS) prospective birth cohort began as a double-blind placebo-controlled randomized controlled trial of the effects of anthelminthic treatment in pregnancy and childhood on immunological and disease outcomes in childhood (ISRCTN32849447), as described elsewhere [19]. Briefly, pregnant women were enrolled into the EMaBS during their first antenatal care clinic visit at Entebbe hospital and randomized to receive either albendazole or placebo and praziquantel or placebo in a 2 × 2 factorial design. Their children formed the EMaBS birth cohort that was followed up from birth with scheduled annual visits when anthropometric measurements and blood samples were taken. At age 15 months, children were randomized to receive either quarterly albendazole or placebo up to five years of age. Maternal health and sociodemographic data were collected at enrolment. Children were actively followed up and tested for malaria and other infections during quarterly visits to the study clinic. Maternal Hb levels were measured at enrolment during the second or third trimester of pregnancy while Hb levels were measured annually in children. Mothers with Hb levels < 8 g/dL (*n* = 54) were treated with albendazole and excluded from the study at enrolment. Biomarkers of iron status were measured at two years of age while child developmental scores were assessed at 15 months and five years using locally adapted tools.

The aims of the current study were: (1) to assess the effect of maternal Hb levels and mild anemia during pregnancy, and Hb levels and moderate anemia in children at one year on developmental scores at 15 months; (2) to assess the effect of maternal and childhood Hb levels and any moderate anemia event at or below five years of age on developmental scores at five years of age; and (3) to assess the effects of ID and IDA at two years of age on developmental scores at five years of age (Figure 1).

### 2.2. Laboratory Procedures and Definitions

Hb levels and mean corpuscular volume (MCV) in mothers and children were measured using the Coulter analyser (Beckman Coulter AC-T 5 diff CP; Beckman Coulter, Nyon, Switzerland). Measures of iron status included plasma ferritin (chemiluminescent microparticle immunoassay [CMI], Abbott Architect, IL, USA), soluble transferrin receptor (sTfR) (Human sTfR ELISA, BioVendor, Brno-Řečkovice, Czech Republic), transferrin (CMI, Abbott Architect, IL, USA), hepcidin (DRG hepcidin 25 [bioactive] High Sensitive enzyme-linked immunosorbent assay [ELISA] kit, DRG Diagnostics, Springfield, NJ, USA) and C-reactive protein (CRP) levels (MULTIGENT CRP Vario assay, Abbott Architect, IL, USA). Inflammation was defined as CRP > 5 mg/L [20], while iron deficiency (ID) was defined using the WHO recommended definition as plasma ferritin < 12 μg/L in the absence of inflammation or <30 μg/L in the presence of inflammation [21]. Mild anemia was defined as Hb < 11 g/dL, moderate anemia as Hb < 10 g/dL and severe anemia as Hb < 7 g/dL after adjusting for altitude (1000 m above sea level) by subtracting 0.2 g/dL from all Hb values [22]. Iron deficiency anemia (IDA) was defined as the presence of both ID and mild anemia. Microcytic anemia was defined as Hb < 10 g/dL and MCV < 80 fL, anemia associated with malaria was defined as Hb < 10 g/dL and malaria parasitemia and anemia associated with stunting was defined as Hb < 10 g/dL with stunting. A clinical malaria episode was defined as parasitemia and temperature > 37.5 °C. Anthropometry data i.e., height, and weight were transformed into height-for-age (HAZ), weight-for-age (WAZ), weight-for-height (WHZ) z-scores using the WHO z-anthro program, and categorised into stunting (HAZ < −2 SD), underweight (WAZ < −2 SD) and wasting (WHZ < −2 SD) respectively [23].

### 2.3. Developmental Assessments

At age 15 months, motor and cognitive abilities were assessed using measures that were developed or translated within the East African region. These included two measures of executive function, the A-not-B task and a self-control (delay inhibition) task, previously translated and used in rural Kenya [24]. Infants’ language, self-care and recognition of self and others were determined using caregiver reports [24]. Fine-motor and gross-motor function were tested using psychomotor ratings in the Kilifi Developmental Inventory [25]. The measures used at age 15 months are summarized in Appendix A. As previously reported, the measures had high inter-rater reliability, and principal components analysis revealed that psychomotor tasks loaded on the same component while cognitive measures loaded on another component [26]. For the current analyses, we grouped the 15-month scores into four domains: [1] executive function, a sum of A not B task and self-control scores; [2] psychomotor, a sum of fine-motor and gross-motor scores; [3] social cognition, a sum of self-care and recognition of self and others; and [4] language scores.

At five years, cognitive and motor assessments were performed using thirteen measures. In the current analyses, we excluded four tests (counting span, running memory, shapes task and tower of London) that were administered to only a small proportion (<20%) of the children. For the remaining nine tests, we used principal components analysis to generate scores for three components of child development: verbal and non-verbal intelligence quotient (IQ), executive function (EF) and motor ability (Appendix A). We further excluded the tap one tap twice task and sentence repetition tests, which did not load well on the IQ, EF, or motor ability components. Details of the translation and validation of all the tests are described elsewhere [27]. In a small pilot study, the tests had good internal consistency (Cronbach’s alpha = 0.65–0.82) and test-retest reliability (correlation coefficient (r) = 0.45–0.88) except the Wisconsin card sort test (r = 0.22) which was administered to only 19 children in the pilot study [27]. The tests used at age five years are summarized in Appendix A.

### 2.4. Statistical Analyses

Statistical analyses were conducted using STATA version 15.1 (StataCorp, College Station, TX, USA). Participant socio-demographic and health characteristics, iron and hemoglobin profiles and developmental scores were described using means and standard deviations or confidence intervals, or median and interquartile ranges for continuous variables and proportions for binary variables. A total household asset score of six was derived using principal components analysis with one indicating lowest and six indicating highest household socioeconomic status.

All developmental scores at 15 months were skewed and were therefore inversely normal transformed. Developmental scores at five years were normally distributed. Univariable linear regression analysis and a *p*-value cut-off <0.05 were used to identify potential confounding factors to include as covariates in further analyses. To examine associations between anemia/ iron status and developmental scores at 15 months and five years we then conducted multivariable linear regression analyses adjusted for the identified confounding factors. In secondary analyses we explored associations between iron profiles corrected for inflammation and malaria using a regression-correction approach proposed by the Biomarkers Reflecting Inflammation and Nutritional Determinants of Anemia (BRINDA) project, which accounts for the linear effects of inflammation and/or malaria on iron biomarkers [5,28] and five-year developmental scores. We were not able to explore the effects of severe anemia on development due to limited power as only 32 (3.4%) children had severe anemia. We were also not able to evaluate the effects of different etiologies of anemia as most children who had anemia associated with malaria or stunting also had microcytic anemia. We performed subgroup analyses to explore the effects of combined maternal and child anemia on developmental scores at 15 months. We further fitted multivariable fractional polynomial models to allow for nonlinear effects of maternal or child Hb levels on developmental scores at 15 months.

## 3. Results

### 3.1. Characteristics of Study Participants

We included 933 mothers with Hb measurements during pregnancy and 933 children with Hb measurements at one year and developmental scores at 15 months in analyses for aim one. A total of 726 children had complete data for analyses assessing associations between Hb levels/any moderate anemia event below five years and developmental scores at five years (aim two), while 530 children had iron profiles at two years and developmental scores at five years and were included in the analyses for aim three. Mean maternal age was 24 years, about half of the women had secondary or tertiary education, and over 50% had two to four children at enrolment (Appendix A). All maternal characteristics were similar for all three analysis groups (Appendix A). Table 1 shows the characteristics of children who were included in each analysis. Approximately half of the children were male. Stunting was more common (13.9% at 12 months, 26.3% at five years) than underweight or wasting in this study population. Malaria was common, with 28% and 59% of children having had at least one episode of clinical malaria in infancy and in the first five years, respectively. Approximately 5% of 12-month-olds and 14% of five-year-olds had asymptomatic malaria parasitemia at annual review. At one year, 2.4% of children had helminth infections; 19.3% had helminthic infections at five years (Table 1).

The prevalence of ID and anemia and the distribution of individual iron markers, Hb and CRP are shown in Table 2. About 36% and 15.8% of mothers had mild or moderate anemia during pregnancy, respectively. In children, the prevalence of mild, moderate, and severe anemia at 12 months of age was 76.3%, 46.7% and 3.4% respectively and decreased to 13.6%, 4.4% and 0.6% at five years of age, respectively. The prevalence of microcytic anemia and anemia associated with stunting or malaria at 12 months was 45.8%, 7.3% and 4.3% respectively and decreased to 2.5%, 1.9% and 1.4% at five years of age, respectively. About 47.1% of children had at least one episode of moderate anemia during the one to five years of follow-up while at age 2 years, almost a third of children had ID and 18.6% had IDA.

### 3.2. Associations between Participant Characteristics and Developmental Scores at 15 Months and Five Years

Median and interquartile ranges for the individual developmental scores at 15 months and 5 years are shown in Appendix A. In multivariable analyses, girls had lower psychomotor scores at 15 months and higher IQ and motor ability scores at five years compared to boys. Malaria parasitemia at annual review was associated with lower IQ at five years. Stunted children had lower psychomotor, social cognition and language scores at 15 months and lower executive function scores at 5 years compared to non-stunted children. Low socioeconomic status was associated with lower language scores at 15 months and lower executive function at five years, while secondary or tertiary maternal education was associated with higher motor scores at 15 months and higher IQ and executive function at five years compared to primary or no maternal education, as previously reported [26,29] (Appendix A).

### 3.3. Anemia during Pregnancy Is Associated with Reduced Psychomotor Scores in Children at 15 Months

In multivariable analyses, lower maternal Hb levels and mild or moderate anemia during pregnancy were independently associated with lower psychomotor scores at 15 months after controlling for infant Hb levels, but were not associated with executive function, social cognition, or language scores (Table 3 and Table 4).

### 3.4. Lower Hemoglobin Levels at 12 Months of Age Are Associated with Reduced Psychomotor and Language Scores at 15 Months

In children, lower Hb levels at 12 months were independently associated with reduced psychomotor and language scores after controlling for maternal Hb levels and other covariates at 15 months in adjusted models (Table 3). Moderate anemia at 12 months was similarly associated with reduced psychomotor scores at 15 months (Table 4). Hb levels and anemia were not associated with executive function or social cognition. We found no evidence of non-linear associations between maternal or child Hb levels and developmental scores at 15 months using fractional polynomial models.

In subgroup analyses, combined maternal anemia in pregnancy and child anemia at 12 months was strongly associated with reduced psychomotor scores at 15 months (Table 4). We found an interaction between maternal and child anemia in predicting psychomotor scores so that the combined effect of both being born to a mother who was anemic during the pregnancy and having anemia in childhood was larger than the product of the individual effects of each condition (β = −0.28; 95% CI: −0.56, −0.01; *p* value = 0.04). We did not find any interactions between maternal and child anemia in predicting other developmental scores.

### 3.5. Hemoglobin Levels and Anemia in Mothers and Children Are Not Associated with Developmental Scores at Five Years

We observed no evidence of association between maternal or child Hb levels, or any moderate anemia event between one to five years of age, and verbal and non-verbal IQ, executive function, or motor ability at five years of age (Table 5, Appendix A).

### 3.6. Iron Status at Two Years Was Not Associated with Developmental Scores at Five Years

We found no evidence of an association between iron status at two years and developmental scores at five years of age in multivariable analyses. In univariable analyses, we observed associations between ID, ferritin and transferrin levels and executive function scores but these effects were not observed after adjusting for potential confounders (Table 6). Similarly, in secondary analyses we observed no associations between iron profiles and developmental scores at five years of age after correcting markers of iron status for inflammation and malaria using the BRINDA approach [28] (Appendix A).

## 4. Discussion

About 29.5 million pre-school children in Sub-Saharan Africa experience developmental delay [1]. Similar to previous findings in the same cohort, we observed associations between sex, household socioeconomic status, maternal education, stunting, malaria parasitemia and developmental scores at fifteen months and five years [26,29]. Anemia and lack of sufficient iron in early childhood are thought to impair the development of motor and cognitive functions. The present study examined the effect of maternal and child anemia on developmental scores at fifteen months and five years and iron deficiency (ID) at two years on developmental scores at five years in a birth cohort of pre-school Ugandan children. About 36% and 15.8% of mothers had mild (Hb < 11 g/dL) and moderate anemia (Hb < 10 g/dL) respectively during pregnancy. At one year of age, 76.3%, 46.7% and 3.4% of children had mild, moderate, and severe anemia, respectively, while at five years 13.6%, 4.4% and 0.6% of children had mild, moderate and severe anemia, respectively. At two years of age, 31.2% and 18.6% of children had ID and IDA, respectively. Low Hb levels and anemia in mothers during pregnancy and at 12 months in children were associated with reduced psychomotor scores at 15 months. Lower Hb levels at 12 months were also associated with reduced language scores at 15 months. However, we did not observe associations between maternal or child Hb levels, or any moderate anemia event below five years and cognitive or motor scores at five years. We also found no evidence that iron profiles at two years predicted developmental scores at five years.

Lower maternal Hb levels and anemia during pregnancy were associated with reduced psychomotor scores at 15 months after controlling for potential confounders including infant Hb levels. Reduced maternal Hb levels in pregnancy may be caused by various factors including ID, infections such as malaria and helminths, and hemodilution. About 50 % of the cases of anemia in pregnancy are attributed to ID [30]. We were not able to investigate the specific effects of maternal ID and IDA during pregnancy as maternal iron status was not measured in this cohort. A small study in China (*n* = 127) reported that infants with ID at birth as measured from cord blood had impaired recognition memory compared to those with sufficient iron status [31]. In contrast to our findings, one study utilizing datasets from two mother-child cohorts in Ghana (*n* = 1023) and Malawi (*n* = 675) reported no associations between maternal Hb or iron status and motor or language development in children at 18 months, which might be explained by the mothers receiving lipid-based nutrient supplementation, including iron and folic acid during pregnancy [16]. A cohort study of Beninese mother-child pairs (*n* = 636) reported an inverted U-shaped association between maternal Hb levels and infant gross motor scores with maternal Hb levels 9–11 g/dL being optimal for gross motor development [17] but did not observe associations between maternal iron status and cognitive, motor and language development [18]. However, we observed that maternal Hb levels above 11 g/dl were associated with improved psychomotor scores and did not identify a non-linear relationship between maternal Hb levels and developmental scores. Similar to our study, they did not observe an association between maternal Hb and language scores at one year [17]. Hemodilution, the disproportionate 50% physiologic increase in blood plasma volume with a 20–30% increase in red cell mass, occurs mostly during the second and third trimesters of pregnancy and is important to improve the transport of oxygen and nutrients to the developing fetus [32]. Low maternal Hb levels may affect child neurodevelopment through effects on fetal physiological functions that affect in utero brain development processes including neuronal migration, myelination, synaptogenesis, and development of the hippocampus, increasing the child’s susceptibility to developmental delay or disability after birth [33,34].

In the current study, we found that low Hb levels and moderate anemia in infancy were associated with reduced psychomotor scores at 15 months of age. Our findings agree with two separate cross-sectional studies (*n* = 1417) that reported associations between anemia (Hb < 10 g/dL), ID, and IDA and poor motor ability in Zanzibari children aged five to 19 months [35,36]. Similarly, one small cross-sectional study (*n* = 102) in Indonesia found that children with anemia (Hb < 11 g/dL) had lower fine motor scores compared to non-anemic children at one to two years of age [37]. In contrast to our findings, a cross-sectional study of six-month-old South African children (*n* = 750) reported no associations between anemia (Hb < 11 g/dL) and psychomotor scores [38]. Fewer children had mild anemia (36.4%) in that study population compared to our study (76.3%) and they did not explore the effects of moderate anemia which may explain the disparity in findings. The combined effects of anemia during pregnancy and in infancy had the largest impact, so that anemia during both time periods resulted in a much larger reduction in psychomotor development. This coincides with the period of rapid brain development, from conception up to the first 1000 days after birth [39]. Exposure to anemia during this period may have detrimental effects on brain processes involved in psychomotor development. Anemia may also affect psychomotor function through associated symptoms such as withdrawal and lethargy.

We similarly observed an association between higher Hb levels at one year and higher language scores at 15 months but found no associations with executive function or social cognition scores. In line with our findings, one study utilizing data from two child cohorts in Malawi (*n* = 1385) and Burkina Faso (*n* = 1122) reported an association between Hb levels at six to nine months and at 18 months and language development at 18 months [16]. In contrast, one cross-sectional study of 226 Egyptian children reported no association between IDA at four to six years and language scores at the same age [40]. In contrast to epidemiological studies, animal studies consistently indicate that iron may influence brain processes involved in language function through its roles in myelination and neurotransmission in the hippocampus and the cerebellum, regions that mediate attention, memory and learning processes [41,42,43]. The causes of anemia are multifactorial among children living in Africa and it is possible that anemia itself or anemia due to a specific cause may impair psychomotor development. In our study, microcytic anemia, which may have indicated ID, was common (46% of children) at 12 months of age and anemia associated with stunting or malaria was less prevalent (7.4% and 4.3%, respectively). However, we were not able to ascertain associations with specific causes of anemia in infancy as most of the children who had anemia associated with stunting or malaria also had microcytic anemia.

We found no associations between Hb levels or moderate anemia below five years and developmental scores at five years despite lower Hb levels influencing development at fifteen months. In contrast with our findings, two large cross-sectional studies of Ecuadorean (*n* = 3153) and Chinese (*n* = 1293) children reported associations between anemia (Hb < 11 g/dL) at three to six years and poor cognitive development at the same age [44,45]. It is likely that the causes of anemia may differ in Ugandan compared to Ecuadorean and Chinese children. Why might Hb levels or anemia influence developmental scores at 15 months but not at five years of age? As previously observed in African children, fewer children were moderately anemic at five years of age (4.4% compared to 46.7% at one year) and numbers were insufficient for subgroup analyses. It is possible that the effects of anemia on psychomotor and language scores observed in infancy improved as anemia resolved with increasing age. Additionally, the differing effects of anemia on development at the two time-points may be due to the influence of changes in risk factors such as household socioeconomic status, nutritional changes and preschool experience.

We found no evidence of association between iron status at two years and cognitive or motor scores at five years in agreement with one cross-sectional study of 541 Ethiopian children which reported no association between ID (serum ferritin < 12μg/L) at 54 to 60 months and cognitive scores at the same age [46]. Since inflammation and malaria alter measures of iron status, we further corrected iron biomarkers for inflammation and malaria using the BRINDA approach, but our findings were similar after this adjustment [28]. Other studies in older children provide mixed evidence for the effects of iron status on cognitive and motor development. One cohort study of 143 Icelandic children reported no associations between iron status at twelve months and six years and cognitive and motor development at six years [47] while a cohort study in Chile (*n* = 163) reported that IDA at 12 to 23 months was associated with impaired cognitive and motor development at five years of age [48]. One cross-sectional study in Turkey (*n* = 172) children reported associations between IDA at six months to six years and motor scores at the same age [49]. It is possible that the effects of ID are most marked during infancy when brain development is more rapid. The lack of association observed in our study may in part be because of the long interval between iron measurement and developmental assessments as the iron status of children may have improved during the three-year interval and the effects of previous deficits may have declined or been overtaken by other factors [5]. Two cross-sectional studies in 1417 Zanzibari children reported associations between ID and IDA at five to nineteen months and poor motor ability at the same age [35,36]. Additionally, children in our study cohort were generally healthy as indicated by the few children with severe anemia at five years. A few randomized controlled trials in African children have yielded conflicting findings with some showing significant associations between iron supplementation and developmental scores [50,51] while others did not find an effect [52,53]. Of the studies that reported beneficial effects, one included mostly anemic children and the other reported benefits in anemic compared to non-anemic children. Improvement of IDA-related symptoms such as withdrawal and lethargy following iron supplementation may result in improved developmental scores.

Despite mixed findings from epidemiological studies, animal and in vitro studies indicate that iron is important for brain development and normal growth, and its deprivation can result in brain dysfunction manifested in poor performance in developmental assessments [54,55,56]. Iron plays an important role in neurotransmission and myelination in parts of the brain including the hippocampus and the basal ganglia that are important for memory, learning processes, language and motor function [9,41,42,57,58]. Animal studies show that CNS iron decreases before restriction in red cell production occurs [56], and behavioral and cognitive function have been observed to improve with iron-replacement therapy, in many instances before an increase in Hb concentration implying that the effects of ID may manifest before IDA manifests [56,59]. However, the application of evidence from animal and in vitro studies to humans may be limited as these studies are conducted in controlled environments with limited confounders and can manipulate iron availability to extents that may not be observed in humans.

A strength of our study is the large sample size and use of both longitudinal and cross-sectional approaches to comprehensively evaluate the effects of maternal and child Hb levels and anemia on developmental scores in children. Different developmental domains were assessed at two time-points using a well-validated extensive battery of measures that were specifically adapted to Ugandan children. Also, data on a wide range of potential confounders were available to adjust for in the multivariable analyses. Our study had some limitations. Our study may not be fully representative of Ugandan children since mothers with severe anemia were excluded from the original trial and all mothers received iron and folate during antenatal care. We also included only about 40% of participants from the original EMaBS birth cohort in our analyses as developmental scores were assessed in smaller subsets of children. However, our study was adequately powered to detect the reported effect estimates. The long interval between iron measurement at two years and scores at five years may have underestimated the true effects of iron (and ID) on developmental scores. Anemia has multiple etiologies, and we were unable to ascertain whether the observed effects of anemia on developmental scores at 15 months were attributable to anemia generally or to specific causes of anemia such as ID because iron status was only assessed at two years. Future studies employing measurement of anemia and iron status and developmental scores together at several time points are recommended.

## 5. Conclusions

Anemia and ID were highly prevalent in our study. We report that low Hb levels during pregnancy and infancy are associated with reduced psychomotor and language scores in Ugandan children at 15 months. These findings emphasize the need to control for anemia in pregnant mothers and young children living in Sub-Saharan Africa where children are exposed to different risk factors such as infections and malnutrition. The current study contributes more evidence to the effects of anemia on early childhood development in Sub-Saharan Africa where such evidence is still scanty and not conclusive. Further well-conducted randomized controlled trials of iron supplementation are required to evaluate the causal effects of iron status on neurobehavioral outcomes in young children living in Sub-Saharan Africa where anemia and ID are common.

## Figures and Tables

**Figure 1 nutrients-14-01452-f001:**
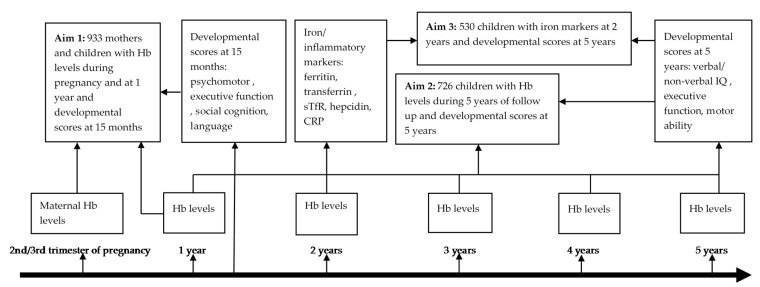
Study design. CRP, C-reactive protein; Hb, hemoglobin levels; sTfR, soluble transferrin receptor.

**Table 1 nutrients-14-01452-t001:** Participant characteristics.

Variable	Analysis for Aim 1; Maternal Hb during Pregnancy, Child Hb 12 Months and Development at 15 Months (*n* = 933)	Analysis for Aim 2; Annual Hb and Development at 5 Years (*n* = 726)	Analysis for Aim 3; Iron Status at 2 Years and Development at 5 Years (*n* = 530)
Age at Hb/iron assessment, mean (SD)	12 months	Annually (1–5 years)	2.4 (0.8) years
Age at developmental assessment, mean (SD)	15.4 (0.5) months	5.0 (0.8) years	5.0 (0.8) years
Sex, male, *n*/total (%)	470/932 (50.4)	355/724 (49.0)	260/530 (49.1)
* Any malaria parasitemia, *n* (%)	46/929 (5.0)	102/726 (14.1)	72/530 (13.7)
** Clinical malaria episodes, *n* (%)			
None	674/932 (72.3)	297/722 (41.2)	216/530 (40.8)
≥1	258/932 (27.7)	425/722 (58.9)	314/530 (59.3)
*** Helminthic infections			
Schistosomiasis, *n*/total (%)	0/877 (0)	28/726 (3.9)	15/530 (2.8)
Hookworm, *n*/total (%)	2/877(0.2)	19/726 (2.6)	13/530 (2.5)
Any worm infection, *n*/total (%)	21/877 (2.4)	140/726 (19.3)	101/530 (19.1)
Treatment with albendazole, *n*/total (%)	462/930 (49.7)	354/724 (48.9)	254/530 (47.9)
Nutritional status at developmental assessment			
Stunting, *n*/total (%)	127/921 (13.8)	187/710 (26.3)	136/520 (26.2)
Underweight, *n*/total (%)	79/932 (8.9)	70/714 (9.8)	59/523 (11.3)
Wasting, *n*/total (%)	39/921 (4.2)	53/713 (7.4)	40/523 (7.7)
Household socioeconomic status, *n*/total (%)			
1 (lowest)	51/916 (5.6)	46/713 (6.5)	30/521 (5.8)
2	68/916 (7.4)	45/713 (6.3)	38/521 (13.1)
3	287/916 (31.3)	219/713 (30.7)	163/521 (31.3)
4	268/916 (29.3)	210/713 (29.5)	143/521 (27.5)
5	189/916 (20.6)	142/713 (19.9)	108/521 (20.7)
6 (highest)	53/916 (5.8)	51/713 (7.2)	39/521 (7.5)

Hb, hemoglobin; SD, standard deviation. Stunting, underweight and wasting were defined as height-for-age Z scores < −2 SD, weight-for-age Z scores < −2 SD and weight-for-height Z scores < −2 SD respectively. Household socioeconomic status was derived using principal components analysis as a composite of building materials of the home, number of rooms and items owned. A total asset score was calculated with 1 indicating lowest and 6 indicating highest household socioeconomic status. * All children were tested annually for asymptomatic malaria parasitemia. ** clinical malaria episodes were recorded for symptomatic children during longitudinal follow-up. *** All children were tested annually for helminthic infections during longitudinal follow-up.

**Table 2 nutrients-14-01452-t002:** Distribution of hemoglobin levels and markers of iron status and inflammation.

Child Anemia/Iron Status	
Anemia	
Moderate anemia at age one, *n*/total (%)	436/933 (46.7)
Moderate anemia at age two, *n*/total (%)	127/615 (20.7)
Moderate anemia at age three, *n*/total (%)	26/641 (4.1)
Moderate anemia at age four, *n*/total (%)	13/399 (3.3)
Moderate anemia at age five, *n*/total (%)	31/713 (4.4)
Any moderate anemia event from one to five years of age, *n*/total (%)	342/726 (47.1)
Any mild anemia event from one to five years of age, *n*/total (%)	550/726 (75.8)
Microcytic anemia at 12 months of age, *n*/total (%)	427/933 (45.8)
Anemia associated with malaria at 12 months of age, *n*/total (%)	40/929 (4.3)
Anemia associated with stunting at 12 months of age, *n*/total (%)	67/921 (7.3)
Annual hemoglobin (Hb) levels	
Hb at age 1 (g/dL), mean (95% CI) (*n* = 933)	9.9 (9.8, 10.0)
Hb at age 2 (g/dL), mean (95% CI) (*n* = 615)	10.9 (10.8, 11.0)
Hb at age 3 (g/dL), mean (95% CI) (*n* = 641)	11.7 (11.6, 11.8)
Hb at age 4 (g/dL), mean (95% CI) (*n* = 399)	12.1 (11.7, 12.6)
Hb at age 5 (g/dL), mean (95% CI) (*n* = 713)	12.0 (11.9, 12.1)
Iron status	
Iron deficiency at 2 years of age, *n*/total (%)	152/490 (31.0)
Iron deficiency anemia at 2 years of age, *n*/total (%)	82/441 (18.6)
Iron and inflammation biomarkers at two years of age	
Ferritin (μg/L), mean (95% CI) (*n* = 495)	21.7 (19.8, 23.7)
Transferrin (g/L), mean (95% CI) (*n* = 517)	2.7 (2.7, 2.8)
sTfR (mg/L) mean (95% CI) (*n* = 518)	6.3 (5.9, 6.7)
Hepcidin (μg/L) mean (95% CI) (*n* = 516)	7.8 (7.0, 8.6)
C-reactive protein (mg/L) mean (95% CI) (*n* = 519)	1.3 (1.2, 1.5)
Maternal Hb/ anemia status at enrolment in pregnancy	
Hb levels (g/dL), mean (95% CI) (*n* = 933)	11.4 (11.3, 11.4)
Mild anemia, *n*/total (%)	336/933 (36.0)
Moderate anemia, *n*/total (%)	147/933 (15.8)

CI, confidence interval; Hb, hemoglobin; SD, standard deviation; sTfR, soluble transferrin receptor. Iron deficiency was defined as ferritin < 12 μg/L in the absence of inflammation or <30 μg/L in the presence of inflammation (CRP > 5 mg/L); Iron deficiency anemia was defined as the presence of both iron deficiency and anemia. Moderate and mild anemia were defined as Hb < 10 g/dL and <11 g/dL, respectively. Microcytic anemia was defined as Hb < 10 g/dL and mean corpuscular volume < 80 fL, anemia associated with malaria was defined as Hb <10 g/dL and malaria parasitemia, anemia associated with stunting was defined as Hb < 10 g/dL and stunting. All Hb measures were adjusted for change in altitude (1000 m above sea level). Mean values are geometric means except for transferrin and Hb which are arithmetic means. Children with missing developmental scores were excluded.

**Table 3 nutrients-14-01452-t003:** Univariable and multivariable linear regression results for the associations between maternal and child hemoglobin levels and developmental scores at 15 months.

Developmental Domain	*n*	Univariable Model β (95% CI)	*p* Value	*n*	Multivariable Model * β (95% CI)	*p* Value
Maternal Hemoglobin Levels and Developmental Scores at 15 Months						
Executive function (A not B + self-control)	686	−0.01 (−0.06, 0.04)	0.71	619	−0.002 (−0.06, 0.05)	0.93
Psychomotor (fine motor + gross motor)	915	0.06 (0.02, 0.11)	0.01	828	0.05 (0.0002, 0.09)	0.05
Social cognition (recognition of self and others + self-care)	933	0.03 (−0.1, 0.07)	0.19	843	0.02 (−0.03, 0.06)	0.50
Language	933	0.02 (−0.02, 0.06)	0.08	843	−0.01 (−0.05, 0.04)	0.81
Child hemoglobin levels at 12 months and developmental scores at 15 months						
Executive function (A not B + self-control)	686	0.01 (−0.04, 0.07)	0.60	619	0.03 (−0.03, 0.09)	0.28
Psychomotor (fine motor + gross motor)	915	0.07 (0.03, 0.12)	0.001	828	0.07 (0.02, 0.12)	0.003
Social cognition (recognition of self and others + self-care)	933	0.03 (−0.4, 0.10)	0.40	843	0.13 (−0.10, 0.36)	0.28
Language	933	0.04 (−0.01, 0.08)	0.09	843	0.05 (0.002, 0.10)	0.04

CI, confidence interval. * Multivariable models were adjusted for age at developmental assessment, sex, stunting, socioeconomic status, maternal education, helminthic infections, and malaria parasitemia. Multivariable models are additionally adjusted for child Hb levels at 12 months or maternal Hb levels during pregnancy. All developmental scores at 15 months were inverse transformed to normalize their distributions.

**Table 4 nutrients-14-01452-t004:** Multivariable linear regression results for associations between maternal and child anemia and developmental scores at 15 months.

Developmental Domain	*n*	Univariable Model β (95% CI)	*p* Value	*n*	Multivariable Model * β (95% CI)	*p* Value
Mild Maternal Anemia during Pregnancy and Developmental Scores at 15 Months (*n* = 336/933)						
Executive function (A not B + self-control)	686	−0.03 (−0.19, 0.12)	0.68	619	0.03 (−0.20, 0.13)	0.68
Psychomotor (fine motor + gross motor)	915	−0.19 (−0.32, −0.05)	0.01	828	−0.14 (−0.27, −0.002)	0.05
Social cognition (recognition of self and others +self-care)	933	−0.04 (−0.17, 0.09)	0.55	843	0.02 (−0.12, 0.16)	0.77
Language	933	−0.01 (−0.14, 0.12)	0.88	843	0.05 (−0.09, 0.19)	0.48
Moderate child anemia at 12 months and developmental scores at 15 months (436/933) ^a^						
Executive function (A not B + self-control)	686	−0.002 (−0.15, 0.15)	0.97	619	−0.03 (−0.20, 0.13)	0.70
Psychomotor (fine motor + gross motor)	915	−0.18 (−0.31, −0.06)	0.01	828	−0.21 (−0.35, −0.08)	0.002
Social cognition (recognition of self and others +self-care)	933	−0.16 (−0.29, −0.03)	0.01	843	−0.59 (−1.24, 0.05)	0.07
Language	933	−0.05 (−0.18, 0.08)	0.45	846	−0.09 (−0.23, 0.05)	0.19
^b^ Combined maternal anemia during pregnancy and child anemia at 12 months and developmental scores at 15 months (*n* = 175/933)						
Executive function (A not B + self-control)	686	−0.08 (−0.27, 0.11)	0.41	619	−0.10 (−0.30, 0.11)	0.36
Psychomotor (fine motor + gross motor)	915	−0.36 (−0.52, −0.19)	0.0002	828	−0.35 (−0.52, −0.19)	0.00004
Social cognition (recognition of self and others +self-care)	933	−0.12 (−0.29, 0.04)	0.14	843	−0.08 (−0.25, 0.10)	0.39
Language	933	0.02 (−0.14, 0.19)	0.78	843	0.03 (−0.14, 0.21)	0.71

CI, confidence interval. Mild and moderate anemia were defined as hemoglobin levels < 11 g/dL and <10 g/dL respectively, adjusting for high altitude (1000 m above sea level). * Multivariable models were adjusted for age at developmental assessment, sex, stunting, socioeconomic status, maternal education, helminthic infections and malaria parasitemia. Multivariable models were additionally adjusted for child Hb levels at 12 months or maternal Hb levels during pregnancy. All developmental scores at 15 months were inverse transformed to normalize their distributions. ^a^ Presented results are for associations with moderate child anemia at 12 months as mild anemia was very common (76.3%) among children at that age. ^b^ Combined anemia is mild maternal anemia during pregnancy and moderate child anemia.

**Table 5 nutrients-14-01452-t005:** Univariable and multivariable linear regression results for the association between hemoglobin levels or anemia and developmental scores at five years.

Developmental Domain	*n*	Univariable Model β (95% CI)	*p* Value	*n*	Multivariable Model * β (95% CI)	*p* Value
Hemoglobin Levels at 5 Years and Developmental Scores at 5 Years						
Verbal and non-verbal IQ	713	0.0002 (−0.06, 0.06)	0.99	681	−0.002 (−0.06, 0.06)	0.96
Executive function	713	−0.04 (−0.10, 0.02)	0.15	681	−0.06 (−0.11, 0.002)	0.06
Motor ability	713	−0.004 (−0.05, 0.05)	0.88	681	−0.001 (−0.05, 0.05)	0.99
Any moderate anemia event below five years and developmental scores at five years						
Verbal and non-verbal IQ	726	−0.18 (−0.31, 0.08)	0.25	693	0.02 (−0.19, 0.22)	0.22
Executive function	726	0.01 (−0.18, 0.20)	0.91	693	0.11 (−0.08, 0.30)	0.25
Motor ability	726	−0.04 (−0.19, 0.12)	0.66	693	0.04 (−0.13, 0.20)	0.67

CI, confidence interval. Moderate anemia was defined as Hb < 10 g/dL adjusting for change in altitude (1000 m above sea level). * The multivariable models were adjusted for age at developmental assessment, sex, stunting, socioeconomic status, maternal education, maternal Hb levels during pregnancy, helminthic infections and malaria parasitemia.

**Table 6 nutrients-14-01452-t006:** Multivariable linear regression results for the association between measures of iron status and anemia at two years of age and cognitive and motor scores at five years of age.

Iron Parameter	*n*	Multivariable Model, β (95% CI)	*p* Value
Verbal/ Non-Verbal IQ			
ID	471	0.09 (−0.16, 0.35)	0.45
IDA	454	−0.02 (−0.37, 0.32)	0.90
Ferritin (μg/L)	471	−0.12 (−0.25, 0.02)	0.09
Transferrin (g/L)	495	0.14 (0.06, 0.33)	0.18
sTfR (mg/L)	490	−0.15 (−0.32, 0.02)	0.09
Hepcidin (μg/L)	489	−0.03 (−0.14, 0.07)	0.54
Executive function			
ID	471	0.21 (−0.04, 0.45)	0.09
IDA	454	−0.11(−0.44, 0.22)	0.53
Ferritin (μg/L)	471	−0.11 (−0.24, 0.02)	0.11
Transferrin (g/L)	495	0.13 (−0.06, 0.32)	0.18
sTfR (mg/L)	490	−0.05 (−0.21, 0.11)	0.54
Hepcidin (μg/L)	489	−0.05 (−0.14, 0.05)	0.32
Motor ability			
ID	471	0.12 (−0.09, 0.34)	0.25
IDA	454	0.09 (−0.19, 0.38)	0.52
Ferritin (μg/L)	471	−0.07 (−0.19, 0.04)	0.20
Transferrin (g/L)	495	0.01 (−0.18, 0.15)	0.87
sTfR(mg/L)	490	−0.07 (−0.21, 0.07)	0.32
Hepcidin (μg/L)	489	−0.04 (−0.12, 0.05)	0.39

CI, confidence interval; sTfR, soluble transferrin receptor; ID, iron deficiency; IDA, iron deficiency anemia. ID was defined as plasma ferritin < 12 μg/L in the absence of inflammation or <30 μg/L in the presence of inflammation (CRP > 5 mg/L); Iron deficiency anemia was defined as the presence of both iron deficiency and mild anemia (Hb < 11 g/dL). All Hb measures were adjusted for change in altitude (1000 m above sea level). The multivariable models were adjusted for age at iron measurement, sex, stunting, inflammation, helminthic infections, socioeconomic status, maternal education, and malaria parasitemia. Ferritin, sTfR, hepcidin and CRP levels were natural log (ln) transformed to normalize their distribution.

## Data Availability

The data presented in this study are available on request from the Principal Investigator of the EMaBS study (A.M.E). The data are not publicly available due to data privacy restrictions.

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
