# Peer review of "Low Hemoglobin Levels Are Associated with Reduced Psychomotor and Language Abilities in Young Ugandan Children"

_nutrients, 2022, doi:10.3390/nu14071452_

Round 1
Reviewer 1 Report
This study consisted of examining the effects of maternal Hb levels as well as child iron status and Hb levels at various ages on developmental scores in preschool children from Uganda. Data from was obtained from the prospective birth cohort Entebbe Mother and Baby Study. Regression analyses were used and adjusted for potential confounders. The authors report that low Hb levels were associated with reduced psychomotor and language performance at 15 months of age, but no associations at 5 years of age.
In general, the study is well designed and executed, and the manuscript is well-written. A few points to consider:
- The authors report animal studies that show developmental ID is linked to behavioural changes that may be reversed with iron supplementation. However, there are also studies that show lasting behavioural consequences in adulthood despite early iron supplementation (PMID: 18424604).
- This study makes the important distinction between the effects of maternal versus child Hb/iron status, and this is an important because the two may not coincide. Although the importance of child iron/Hb status on these outcomes is evident, It would be helpful if the authors could explain why maternal Hb is important in predicting outcomes in the child.
- Related to the point 2, another critical aspect is distinguishing between the acute effects of iron deficiency and anemia (which are likely reversible) versus the lasting developmental consequences of iron deficiency and anemia (which are likely irreversible). Although the authors make an effort to distinguish the two in some sections, this aspect could be clarified in others. For example, when discussing the literature linking iron status and anemia with behavioural outcomes (pg 11, lines 335-384), it would be helpful if the authors specified whether these hematological or biomarker assessments were done at the time of testing (ie is iron deficiency or anemia present at the time of behavioural testing?). In this regard, perhaps the lack of association between iron status/Hb levels and cognitive/motor scores at 5 years reflects the low prevalence of anemia in these children (i.e. the behavioural changes seen at 15 months reflect acute effects of anemia/iron deficiency, and since anemia is no longer prevalent at 5 years of age, these behavioural outcomes are no longer evident)?
- Figure 1 is somewhat confusing. Why are Hb levels at 3 yr going right into aim 2, instead of merging with all Hb arrows across the years?
Reviewer 2 Report
This is an interesting and well performed epidemiologic study, providing additional information on the complex interaction of maternal and child anemia and iron deficiency on development.
1. The authors should comment on the phenomenon of "physiologic anemia of pregnancy", where hemoglobin concentration is decreased due to expansion of maternal plasma volume despite increased red cell production. In the second trimester particularly, a hemoglobin that would indicate moderate anemia in this study would be considered a normal variant.
2. Do the authors have access to data on maternal iron status during pregnancy? It would be interesting to include this if so. If the data are not available or if that additional analysis is outside the scope of the study as approved, the authors should comment on what that might add to the understanding of the topic.
